# Chorioretinitis among Immigrant and Travellers. Comment on Mansour et al. Presumed Onchocerciasis Chorioretinitis Spilling over into North America, Europe and Middle East. *Diagnostics* 2023, *13*, 3626

**DOI:** 10.3390/diagnostics14050478

**Published:** 2024-02-23

**Authors:** Juliet Otiti-Sengeri, Kenneth Lado Lino Sube, Joseph Nelson Siewe Fodjo, Kenneth Bentum Otabil, Robert Colebunders

**Affiliations:** 1Department of Ophthalmology, Makerere University, Kampala P.O. Box 7072, Uganda; ojsjuliet@hotmail.com; 2Department of Ophthalmology, Juba of University, Juba P.O. Box 82, Sudan; ladolojuan@gmail.com; 3Global Health Institute, University of Antwerp, 2610 Antwerp, Belgium; josephnelson.siewefodjo@uantwerpen.be; 4Department of Biological Science, University of Energy and Natural Resources, Sunyani P.O. Box 214, Ghana; kenneth.otabil@uenr.edu.gh; 5Department of Tropical Disease Biology, Liverpool School of Tropical Medicine, Liverpool L3 5QA, UK

We read, with interest, the paper by Mansour et al. describing four patients with presumed onchocerciasis chorioretinitis [1]. Onchocerciasis is a neglected tropical disease caused by the nematode *Onchocerca volvulus*, transmitted by the bites of female-infected blackflies (*Simulium*) [2]. Globally, an estimated 20.9 million people are infected with *O. volvulus*, with 99% of them living in sub-Saharan Africa [2]. Onchocerciasis is well known to cause skin and eye disease (river blindness) [2]. From 1970 to 2020, the number of international refugees, immigrants, and migrants tripled, reaching an estimated 281 million [3]. As some may originate from onchocerciasis-endemic regions, it is vital for clinicians in non-endemic regions to recognize and diagnose onchocerciasis, including ocular manifestations [4].

Most *O. volvulus* infections diagnosed outside of endemic areas occur in immigrants, refugees, and travellers, almost exclusively from sub-Saharan Africa [5]. Imported onchocerciasis among migrant populations is likely to be underdiagnosed [4]. For instance, in a single Spanish center between 1989 and 2007, almost 400 cases of onchodermatitis were observed in persons originating from Equatorial Guinea [6]. Active screening for *O. volvulus* infection increases the number of infected people, as shown by a study in Israel in which clinicians diagnosed onchocerciasis in 83 Ethiopian migrants through the active referral and screening of people with pruritis or rash. None of these cases were detected during a general health examination upon arrival in Israel [7].

Onchocerciasis predominantly prevails in very remote areas. Thus, inhabitants of these areas may have fewer opportunities to migrate. Moreover, since the introduction of community-directed treatment with ivermectin (CDTI), *O. volvulus* transmission decreased in most onchocerciasis-endemic foci [2]. Regular annual ivermectin treatment can successfully eliminate *O. volvulus* microfilariae from the skin and/or prevent their migration into the anterior eye chamber, thereby averting new-onset ocular pathology [8]. Ivermectin also has a therapeutic effect early in the course of onchocercal eye disease; regular intake can completely resolve cornea keratitis punctata lesions, while early-stage sclerosing keratitis and iridocyclitis can regress considerably [8]. However, advanced lesions of the anterior and posterior eye segments remain progressive despite ivermectin treatment [9].

Recent trends indicate a decrease in the incidence of imported *O. volvulus* infections, likely due to the success of onchocerciasis elimination programs [6]. From 1994 to 2018, at the travel clinic of the Institute of Tropical Medicine in Antwerp, Belgium, 33 cases of onchocerciasis were detected, with declining numbers over time [10].

In contrast to immigrants, travellers visiting onchocerciasis-endemic areas are rarely diagnosed with onchocerciasis [11,12,13]. Like migrants, they also acquire their infection in sub-Saharan Africa [4,5,10]. Most travellers infected with *O. volvulus* resided for at least three months in endemic areas, with a median duration of approximately two years [6,12,14]. However, onchocerciasis can also be acquired through shorter but intense exposures to highly endemic areas [5,15,16].

Onchocercal eye disease primarily affects migrants from endemic areas and is seldom observed in travellers [12,17]. The prevalence of eye disease among migrants varies, influenced by factors such as *O. volvulus* strain and host genetics [18]. Corneal scarring is the most common ocular pathology. In imported cases, ocular involvement is often mild, although around 10% experience some degree of functional vision loss. Posterior segment eye disease is rarely observed [7]. In the past, a prevalence of eye disease in expatriates living in endemic areas was observed, similar to and even exceeding that in migrants [19,20]. These expatriates often worked as missionaries in remote, highly endemic settings and developed chronic untreated infections.

The clinical manifestation of imported *O. volvulus* infection differs between immigrants from endemic areas and travellers only visiting these areas. Pruritus is the most common symptom in both groups, affecting 60% to 90% of infected individuals [12,17,21]. In travellers, the most frequent dermatologic manifestation is acute papular onchodermatitis [7,17,21,22], while immigrants from endemic areas commonly exhibit chronic forms of skin disease such as chronic papular onchodermatitis, lichenified onchodermatitis, depigmentation, and skin atrophy [12,21,22]. Subcutaneous onchocerciasis nodules are often present in migrants. For example, they were observed in 29% of immigrants to Spain but are uncommon in travellers [21]. Acute fixed unilateral limb oedema is a unique manifestation in travellers, occurring in up to 12% of temporary residents [17,23]. Up to 20% of patients with imported onchocerciasis are asymptomatic, often identified through elevated eosinophil counts [21].

Most migrants with *O. volvulus* infection present symptoms upon immigration, yet in some, these symptoms only appear months to years after moving to non-endemic regions [7,22]. The median time to symptom onset after leaving endemic areas is approximately 1–1.5 years, reflecting the prolonged prepatent period of *O. volvulus* [12]. In the absence of systematic screening, diagnostic delays were observed ranging from 1 to 2 years [7,21].

We acknowledge that the authors considered the diagnosis of their four patients as “presumed onchocerciasis chorioretinitis”. Indeed, there was no definitive proof of onchocerciasis in any of the cases. None underwent testing confirming an *O. volvulus* infection. We understand that in the Middle Eastern clinics visited by the patients, an *O. volvulus* PCR test was unavailable. However, there are many other onchocerciasis diagnostic tests that could have been considered. An excellent method to detect active infection is the detection of *O. volvulus* microfilariae in skin snips through microscopy [24]. Skin snips (skin biopsies) can be obtained by scalpel or surgical blade, by shaving off a small elevated cone of skin (3 mm diameter), by needle, or by a sclerocorneal punch biopsy. Biopsy samples are incubated in saline at room temperature for 24 h to allow microfilariae to emerge from the skin into the saline solution. Skin snips are usually obtained from the iliac crest (hip area), the shoulder blade (scapula), or the lower extremities, areas commonly exposed to the bites of blackflies. Although this method has excellent sensitivity in ivermectin-naïve settings, its sensitivity is reduced after ivermectin treatment. Thiele et al. demonstrated that in an area with a successful CDTI program, skin snip testing by microscopy had a sensitivity of 29% in Ethiopia and 76% in Uganda when compared with PCR, considered a reference/gold standard technique [25].

Mansour et al. mentioned that in the absence of superficial skin lesions, snip biopsies were not performed [1]. It is important to point out that performing two skin snips of normal skin also generally reveals the presence of microfilariae in infected individuals. Moreover, performing four snips improves sensitivity by 1.5- to 2-fold [26].

Slit lamp examinations of the eyes can be used to visualize the anterior chamber of the eye for the presence of microfilariae [9,24].

Another method for the paraclinical diagnosis of onchocerciasis is the detection of *O. volvulus* antibodies using an OV16 IgG4 Enzyme-Linked Immunosorbent Assay (ELISA) test. Such a test is easy to conduct, but it may be difficult to obtain the reagents for the ELISA test. There is also an OV16 rapid diagnostic test (RDT) that can be performed on the spot. The SD Bioline Onchocerciasis IgG4 RDT (Standard Diagnostics [now Abbott], Gyeonggi-do, the Republic of Korea) has a specificity that ranges from 95% to 99% and a sensitivity of approximately 80% [27,28,29,30]. However, it may be difficult to obtain the test if it is not registered in your country. The combined use of three groups of recombinant antigens in conventional ELISA can increase the sensitivity and specificity to reach about 100% for the diagnosis of onchocerciasis [31]. As the authors sought to demonstrate that the observed chorioretinitis was linked to onchocerciasis, at the least, the Ov16 RDT or ELISA that demonstrates prior exposure to *O. volvulus* parasites was necessary to test each case.

Another diagnostic platform integrates four recombinant antigens into a rapid, high-throughput luciferase immunoprecipitation system assay that is 100% sensitive and 80–90% specific in distinguishing onchocerciasis from related filarial infections [27,29,32].

In addition to the lack of diagnostic evidence, there are other reasons to question the diagnosis of onchocerciasis chorioretinitis. None of the four patients took ivermectin before being diagnosed with chorioretinitis. So, if they were infected with *O. volvulus*, they would be expected to have a history of skin lesions and itching. Presumed ocular involvement in onchocerciasis should be supported with a history of itching and/or evidence of typical onchocerciasis skin lesions and nodules. The initial history of swelling and redness of the eyes reported in case 1 is not typical of ocular onchocerciasis. Ocular onchocerciasis is generally a relatively quiet and progressive disease; pain, redness, and eye/facial swelling in such patients would typically present as an immune reaction in response to ivermectin treatment [32].

Although ocular findings in onchocerciasis can sometimes present with posterior segment lesions alone [33], it is unclear whether a slit lamp examination of the anterior segment was performed. In the absence of previous ivermectin treatment, the lack of microfilaria in the anterior chamber or corneal lesions (punctuate keratitis) weakens the argument supporting the diagnosis of an onchocerciasis-related eye disease.

Additionally, the response of uveitis to ivermectin without mentioning any side effects, as reported in case 1, is not typical for a person with an active *O. volvulus* infection who never received treatment. Ivermectin induces the death of the microfilariae in the skin and eyes [33]. Therefore, in an ivermectin-naïve *O. volvulus*-infected individual, ivermectin treatment generally would cause itching and an initial worsening of the uveitis [32].

Living in close proximity to a rapidly flowing river in an onchocerciasis endemic area is indeed a crucial criterion for considering the diagnosis of onchocerciasis [34]. However, developing onchocerciasis eye disease typically requires prolonged and heavy exposure to bites of *O. volvulus*-infected blackflies [24]. Consequently, onchocerciasis predominantly affects farmers who cultivate land adjacent to rivers and also fishermen [35]. From the four case reports, it is not explicitly clear whether there was such extensive exposure to blackflies. The authors mentioned that the patients frequented a nearby river, but the specific name of the river is only provided in the case report of a white Middle Eastern man who resided for three decades near swamps linked to the Wouri River in Cameroon. Although this river is located in an onchocerciasis-endemic area and may serve as a blackfly breeding site, it is worth noting that swamps themselves are not usually breeding sites for blackflies. Additionally, only one patient was a black African woman residing near the riverside in Sierra Leone.

We agree with the authors that there is a lack of diagnostic images for ocular onchocerciasis. However, photographs of posterior segment eye lesions caused by onchocerciasis can be found in the publication by Semba et al. [36]. Various infectious organisms are able to induce neuro-retinitis [37]. Potential differential diagnoses for the four presumably onchocerciasis chorioretinitis cases to be considered include sarcoidosis, toxocariasis [38], viral infections [37], and non-O. volvulus nematode infections [39,40].

There are several points that we would like to address from the authors’ statements. It is acknowledged that the authors highlighted the potential association between onchocerciasis and epilepsy, including nodding syndrome. Recent epidemiological studies provided strong epidemiological evidence for the association between onchocerciasis and epilepsy [41]. This form of epilepsy is now termed onchocerciasis-associated epilepsy (OAE) [42,43], with nodding syndrome as one of its clinical presentations [44]. In contrast to the 10,000 figure mentioned by Mansour et al., recent epidemiological studies indicate that around 300,000 individuals may be affected by OAE [45].

Additionally, the authors mention two main strains of *O. volvulus*, the savanna strain, causing ocular disease, and the rainforest strain, initially believed not to cause blindness despite high parasite burdens. However, upon the reanalysis of studies from West Africa with adjustments for sample size, no statistically significant differences in blindness prevalence were found between forest or savannah habitats [46].

In conclusion, in this era of globalization with the increasing trends in migration and international travel, creating awareness about onchocerciasis among healthcare workers worldwide becomes paramount. However, the diagnosis and management of imported cases of onchocerciasis still pose challenges due to the nonspecific clinical presentations, delayed onset of symptoms, inadequate awareness among healthcare providers in non-endemic regions, and barriers in performing *O. volvulus* diagnostic tests. These challenges may require adjustments in the clinical training curriculum and health policies in non-endemic countries.

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
