# Peer review of "Chorioretinitis among Immigrant and Travellers. Comment on Mansour et al. Presumed Onchocerciasis Chorioretinitis Spilling over into North America, Europe and Middle East. Diagnostics 2023, 13, 3626"

_diagnostics, 2024, doi:10.3390/diagnostics14050478_

Round 1

Reviewer 1 Report

Comments and Suggestions for Authors

I appreciated authors’ effort in highlighting the controversies in the paper of Mansour et al. This allows the potential readers to better understand the diagnosing of ocular onchocerciasis.

I would recommend commenting the problem of the lack of the diagnostic images for ocular onchocerciasis and if the authors have good examples to present some of them in their manuscript.

I’m also not sure that the title of the comment perfectly reflects its main content, which mostly covers diagnosing of onchocerciasis. Please consider modification of the title.

It would also be useful to briefly discuss potential differential diagnosis for the cases presented in the original paper.

Author Response

We thank the reviewer for the excellent comments. Here are our responses. We also revised the paper accordingly.

Reviewer

I appreciated authors’ effort in highlighting the controversies in the paper of Mansour et al. This allows the potential readers to better understand the diagnosing of ocular onchocerciasis.

I would recommend commenting the problem of the lack of the diagnostic images for ocular onchocerciasis and if the authors have good examples to present some of them in their manuscript.

Response

We now state in the paper “We agree with the authors that there is a lack of diagnostic images for ocular onchocerciasis. However, photographs of posterior segment eye lesions caused by onchocerciasis can be found in the publication of Semba et al [36].” We also include the ref of Semba et al.

  1. Semba, R.D.; Murphy, R.P.; Newland, H.S.; Awadzi, K.; Greene, B.M.; Taylor, H.R. Longitudinal study of lesions of the posterior segment in onchocerciasis. Ophthalmology 1990, 97, 1334-1341, doi:10.1016/s0161-6420(90)32413-2.

We also include a picture of onchocerciasis chorioretinitis obtained from WHO/TDR

Reviewer

I’m also not sure that the title of the comment perfectly reflects its main content, which mostly covers diagnosing of onchocerciasis. Please consider modification of the title.

Response

We now changed the title to “The diagnosis of onchocerciasis chorioretinitis among immigrant and travelers”

Reviewer

It would also be useful to briefly discuss potential differential diagnosis for the cases presented in the original paper.

Response

We now state: “Various infectious organisms are able to induce neuro-retinitis [37]. Potential differential diagnoses for the four presumably onchocerciasis chorioretinitis cases to be considered include sarcoidosis, toxocariasis [38], viral infections [37], non-O. volvulus nematode infections [39,40].

We also included the references

  1. Choi, S.K.; Byon, I.S.; Kwon, H.J.; Park, S.W. Case series of neuroretinitis in Korea. BMC Ophthalmol 2024, 24, 24, doi:10.1186/s12886-024-03290-3.
  2. Jee, D.; Kim, K.S.; Lee, W.K.; Kim, W.; Jeon, S. Clinical Features of Ocular Toxocariasis in Adult Korean Patients. Ocul Immunol Inflamm 2016, 24, 207-216, doi:10.3109/09273948.2014.994783.
  3. de Souza, E.C.; Abujamra, S.; Nakashima, Y.; Gass, J.D. Diffuse bilateral subacute neuroretinitis: first patient with documented nematodes in both eyes. Arch Ophthalmol 1999, 117, 1349-1351, doi:10.1001/archopht.117.10.1349.
  4. Bird, A.C.; Smith, J.L.; Curtin, V.T. Nematode optic neuritis. Am J Ophthalmol 1970, 69, 72-77, doi:10.1016/0002-9394(70)91858-1.